| Open Peer Review | Pathogenesis and Host Response | Methods and Protocols

# Evaluation of hepatitis A virus recombinant proteins for detecting anti-HAV IgM and IgG antibodies

Supriya Hunderkar,[1] Nital Ganorkar,[1] Atul Walimbe,[1] Kavita Lole[1]

**ABSTRACT** Hepatitis A virus (HAV) is a major causative agent of self-limiting liver infections. India was highly endemic for HAV in the past; children were exposed to the virus at an early age without any disease symptoms and developed lifelong immunity. With improvements in living conditions, an epidemiological transition is occurring. There is a significant increase in hepatitis A outbreaks involving adolescents and young adults. The gold standard for hepatitis A diagnosis is anti-HAV IgM antibodies. Although antibody responses are primarily targeted against HAV structural proteins (capsid proteins), non-structural proteins are also immunogenic. In the present study, we expressed HAV capsid proteins VP1-2A, VP0 (VP4 + VP2), VP3, and non-structural protein 3C$^{Pro}$ in the bacterial system and explored the possible use of these as antigens to detect anti-HAV IgM and IgG antibodies using a well-defined serum sample panel. The capsid protein-based assays showed overall less sensitivity for detection of both anti-HAV IgM and IgG antibodies as compared to whole virus antigen-based commercial assays. Among capsid proteins, rVP1-2A showed the highest sensitivity (86.3%) and specificity (84.2%) in detecting anti-HAV IgG, while rVP0 (VP2 + VP4) exhibited the highest sensitivity (79.5%) and specificity (80.2%) for IgM antibodies. Interestingly, r3C$^{Pro}$ exhibited higher sensitivity (96.9%) and specificity (93.2%) in IgM detection and 93.94% sensitivity and 88% specificity for IgG, indicating its usefulness in detecting both anti-HAV IgM and IgG antibodies during the acute phase of the disease. Though 3C$^{Pro}$ appeared to be useful in differentiating antibody responses due to infection and vaccination, our analysis revealed that the anti-3C$^{Pro}$ antibody response is short-lived after natural infection, and hence, it cannot be used as a marker to differentiate between infection and vaccination. However, 3C$^{Pro}$ would be useful for developing a hepatitis A diagnostic assay.

**IMPORTANCE** Hepatitis A was highly endemic in India earlier. With recent developments, there is a shift in the endemicity to intermediate levels. This has resulted in the occurrence of hepatitis outbreaks with symptomatic infections in adolescents and adults. Occasionally, the disease manifestations are serious, leading to acute liver failure. In such a situation, there is a need for a timely diagnosis of the infection.

**KEYWORDS** hepatitis A virus, diagnosis, ELISA, capsid proteins, non-structural proteins, viral protease

Hepatitis A virus (HAV) is a major causative agent of self-limiting liver infections. The virus is transmitted through contaminated water and food via the fecal-oral route. Hepatitis A prevalence is associated with the water quality, hygiene, sanitation, and socioeconomic status of the population. In 2019, global burden of disease data estimated 159 million acute infections and 39,000 deaths due to HAV. Among these, 66% of acute hepatitis A cases and 97% of deaths were from low-income and middle-income countries. The Southeast Asia region had the greatest number of estimated cases

Address correspondence to Kavita Lole, lolekavita37@yahoo.com.

The authors declare no conflict of interest.

See the funding table on p. 11.

(42 million) and deaths (24,000), i.e., 60% of the total number of deaths (1). India was highly endemic for HAV in the past; children were exposed to the virus without any disease symptoms and developed lifelong immunity. Typically, 90% seropositivity was seen by the age of 5 years and almost 100% by 10 years. Overall, symptomatic infections were low, and hepatitis A outbreaks were not common. With improvements in living conditions, an epidemiological transition is occurring, and the Indian population is no longer homogeneous for its HAV exposure profile (2, 3). There is a significant increase in hepatitis A outbreaks and symptomatic hepatitis A cases among older children and young adults in different parts of the country (4). It appears that India will soon require the introduction of childhood HAV vaccinations to reduce symptomatic infections and related morbidities.

HAV belongs to the family *Picornaviridae* and the genus *Hepatovirus*. The HAV genome is a single-stranded positive-sense RNA of ~7.5 kb. It encodes a single polyprotein containing ~2,230 amino acids and is divided into three regions: P1 (encoding structural proteins [SPs]: VP1, VP2, VP3, and VP4), P2 (encoding 2B and 2C proteins), and P3 (encoding six non-structural proteins [NSPs]: 2B, 2C, 3A, 3B, 3C$^{Pro}$, and 3DPol). 3C$^{Pro}$ is a cysteine protease that proteolytically processes HAV polyprotein into functional subunits (5). HAV replicates silently in hepatocytes with the release of new virions in a non-lytic way as membrane-wrapped (quasi-enveloped) particles (6).

Laboratory diagnosis of hepatitis A is done by testing serum anti-HAV IgM antibodies, which remain detectable up to 3–6 months after infection. Although antibody responses are mainly targeted against structural proteins VP1, VP2, and VP3 (7), HAV NSPs are also antigenically reactive (8, 9). The presence of antibodies against 3C$^{Pro}$ has been documented in experimentally infected chimpanzees, primates, and infected children (9, 10), while those against P2 region proteins have been shown to remain stable up to a decade after the infection (9). The presence of 19 antigenic domains within the structural protein-encoding region has been demonstrated using synthetic polypeptides (11). These findings indicate the possible use of linear epitopes in SPs for the detection of antibodies against HAV, either produced in response to HAV infection or vaccination. However, as the major antigenic site on HAV capsid is non-linear, the use of recombinant capsid protein antigen is not so effective; commercially available HAV antibody testing kits largely utilize inactivated whole virus as antigen. Such tests specifically detect antibodies against only SPs and not NSPs. Hence, these are not useful to differentiate between immunity acquired by natural infection and vaccination. Furthermore, the growth of HAV in cell culture is slow, and the production of antigens in large quantities is not cost-effective (5).

In view of these limitations, we decided to explore the possible use of recombinant HAV antigens, 3C$^{Pro}$, and capsid proteins, VP1-2A, VP0 (VP4 + VP2), and VP3 expressed in bacterial systems to analyze anti-HAV IgM and IgG antibody responses. For that, we used well-defined antibody panels constituted by serum samples from acute hepatitis A cases, individuals having past exposure to the virus, vaccinated individuals, and patients diagnosed with infection with other hepatitis or picornaviruses.

## MATERIALS AND METHODS

### Serum sample panels

Serum samples collected during hepatitis A outbreaks were screened for anti-HAV IgM antibodies using Wantai anti-HAV IgM kit (Beijing Wantai Biological Pharmacy Enterprise Co. Ltd., Beijing, China) and for IgG antibodies using HEPAVASE A-96 (TMB) kit (General Biologicals Corp., Hsinchu, Taiwan).

Panels of human serum samples were as follows:

1.  Anti-HAV IgG and anti-HAV IgM antibody negative samples ($n = 100$).

2. Anti-HAV IgG antibody positive serum samples.
   a. Mixed-age group ranging from 2 to 60 years ($n = 33$).
   b. Different age groups: 6–10 years, 15–25 years, and 40+ years ($n = 30$ each).
   c. Paired anti-HAV IgG negative and positive samples ($n = 30$ each), from children (age group 2–14 years) who were found to be anti-HAV IgG negative during investigation of hepatitis A outbreak in children in 2004 in Daund, Pune, India (2). These children were vaccinated with two doses of HAV vaccine HAVRIX (GlaxoSmithKline Inc.) with a gap of 8 months. Two weeks after the second dose, blood samples were tested for anti-HAV IgG antibody titers.

3. Symptomatic individuals testing positive for anti-HAV IgM antibodies and having elevated alanine amino transferase levels (>40 IU/L, range: 60 to >150 IU/L) (mixed-age group: 2–12 years) ($n = 39$).

4. Hepatitis B virus (HBV) anti-HBcAg IgM positive, anti-HEV (hepatitis E virus) IgM positive, hepatitis C virus (HCV) RNA positive and Enterovirus (Hand, Foot, and Mouth Disease virus) RNA positive samples, $n = 30$ for each category.

## Cloning and expression of HAV 3C$^{Pro}$ and capsid proteins

The 3C$^{Pro}$ and capsid protein encoding HAV genomic segments were amplified by reverse transcription PCR from a stool sample of an individual infected with genotype IIIA hepatitis A virus. The forward and reverse primer pair had a start and a stop codon, respectively. The primers used for amplification are listed in Table 1.

Amplicons were cloned into a pET28a vector (Invitrogen, Carlsbad, CA, USA) along with an N-terminal His-tag. The insertion was confirmed by sequencing using Big-Dye terminator cycle sequencing ready reaction kit (Applied Biosystems, California, USA). RIPL strain of *Escherichia coli* expression host cells (Invitrogen, Carlsbad, CA, USA) were transformed with pET-His-3C$^{Pro}$, pET-His-VP1-pX, pET-His-VP0, and pET-His-VP3 plasmids separately and used for protein expression. The induction of rVP1-2A, rVP0, and rVP3 proteins was carried out using 1.0 mM isopropyl-d-thiogalactopyranoside (IPTG) at 37°C for 4 h, while r3C$^{Pro}$ was induced using 0.5 mM IPTG for 4 h at 37°C. All proteins were purified at the first level using nickel-nitrilotriacetic acid (Ni-NTA) metal affinity chromatography resin (Invitrogen, Carlsbad, CA, USA) as per the manufacturer's instructions. The r3C$^{Pro}$ protein was purified under native conditions, elution was done with buffer containing 150 mM Imidazole, and fractions were analyzed on 10% SDS-PAGE. Fractions containing protein of expected size were pooled and processed for further purification using size exclusion chromatography (Superdex 200 column, GE Healthcare). The capsid proteins rVP1-pX, rVP0, and rVP3 were purified under denaturing conditions as these proteins largely remained in the insoluble fraction using Ni-NTA metal affinity chromatography. For that, cells were lysed in guanidium lysis buffer, and fractions were collected using buffer containing 8 M urea at pH 4.0 and analyzed on 10% SDS-PAGE. Protein fractions showing expected protein band size were pooled and stepwise dialyzed against buffers containing 6 M, 4 M, and 2 M urea (2 h in each buffer) to remove urea. Protein

**TABLE 1** List of primers used for amplification of HAV genomic segments[a]

| S. no. | Gene | Primer sequence 5′ to 3′ |
| --- | --- | --- |
| 1 | VP0F | GAT GAA TTC ATG GAT GTG GAG GAG GAA CAA ATG ATT |
| 2 | VP0R | GAT AAG CTT TTA C TGT GTA GAC AGG GGT GTT AGC CCA |
| 3 | VP3F | GAT GAA TTC ATG ATG AGA AAT GAA TTT AGA GTC AGT AC |
| 4 | VP3R | GAT AAG CTT TTA C TGA GTT GTG ACA TCC ATA GCA TGA |
| 5 | PxF | GAT GAA TTC ATG GTT GGA GAT GAT TCT GGA GGC TTC |
| 6 | PxR | GAT AAG CTT TTA TTT TTC TTT TAT CTC CTG TAT CCC T |
| 7 | 3C F | GAT CTC GAG ATG TCA ACT TTA GAA ATT GCT GGT TT |
| 8 | 3C R | GAT AAG CTT TTA CTG ACT TTC AAC TAT TTT CTT ATC T |

[a]Underlined sequences are restriction sites used for cloning.

concentration was estimated using the Folin Lowry method. Protein purity was assessed by 10% SDS-PAGE followed by western blot analysis using anti-His antibodies (Sigma-Aldrich).

## Evaluation of reactivity of different HAV proteins with anti-HAV IgM and IgG antibody positive serum samples in indirect ELISA

Before proceeding with testing of serum samples in enzyme-linked immunosorbent assay (ELISA), conditions for ELISA were optimized using different concentrations of HAV proteins for coating, dilutions of anti-HAV IgM and IgG antibody positive serum samples, and anti-human IgM-horseradish peroxidase (HRP) and IgG-HRP conjugates by carrying out checkerboard titrations. Table 2 lists different parameters that were optimized for each HAV recombinant antigen-based ELISA. Broadly, the protocol followed for each ELISA was the coating of antigen in Nunc Maxisorp 96-well plates using an appropriate concentration in 50 mM carbonate buffer (pH 9.2) at 37°C overnight. The next day, wells were blocked with 100 µL of 1× StabilGuard Immunoassay BSA-free solution (Surmodics, USA) at room temperature for 1 h, followed by three washes with 1× phosphate-buffered saline (pH 7.4) containing 0.1% Tween-20 (1× PBST). Human serum samples were diluted (1:100) in a diluent (1× PBS containing 4% non-fat milk powder), added into wells, and incubated at 37°C for 1 h. Following five washes with 1× PBST, HRP-conjugated goat anti-human IgG or anti-human IgM antibodies (Sigma Chemicals, St. Louis, MO, USA) were added and incubated for 30 min at 37°C. The substrate TMB (3,3′,5,5′-tetra-methylbenzidine) was added, the enzymatic reaction was stopped by 4 N $H_2SO_4$, and absorbance was measured at 450 nm in an ELISA reader (BioTek Powerwave 340). Known anti-HAV IgG/IgM antibody positive and negative human serum samples (three each) were included in each ELISA as positive and negative controls.

### Statistical analyses

The cutoff value for each assay was determined by the analysis of the receiver operating characteristic (ROC) curve based on the values of positive samples divided by negative samples (P/N) using the Jamovi project (2023) (jamovi, version 2.3). Similarly, the sensitivity and specificity for each assay were estimated.

## RESULTS

### Expression and purification of recombinant capsid and 3C^Pro proteins

The conditions for protein expression were optimized by using ELISA as a screening assay using known anti-HAV IgM and IgG negative and positive human serum samples. Cells were grown in 500 mL cultures, harvested cell pellets were processed for protein purification, and purified proteins were quantitated. All recombinant proteins were evaluated in ELISA for their reactivity with known serum samples, visualized on SDS-PAGE, and confirmed by western blot analysis with anti-His antibodies (Fig. 1A through F).

### Testing of recombinant capsid and r3C^Pro proteins in indirect ELISA

After confirmation of protein size and purity, each protein was used at different concentrations for coating ELISA plate wells, and conditions for both IgM and IgG ELISAs were

**TABLE 2** Optimum ELISA conditions determined using checkerboard assays

| HAV protein | Coating of antigen/well IgG ELISA (µg) | Dilution of anti-human IgG-HRP conjugate | Coating of antigen/well IgM ELISA (µg) | Dilution of anti-human IgM-HRP conjugate |
|---|---|---|---|---|
| rVP1-pX | 1.0 | 1:10,000 | 1.0 | 1:30,000 |
| rVP0 | 0.5 | 1:15,000 | 1.0 | 1:30,000 |
| rVP3 | 0.5 | 1:10,000 | 1.0 | 1:30,000 |
| r3C^Pro | 1.0 | 1:10,000 | 1.0 | 1:10,000 |

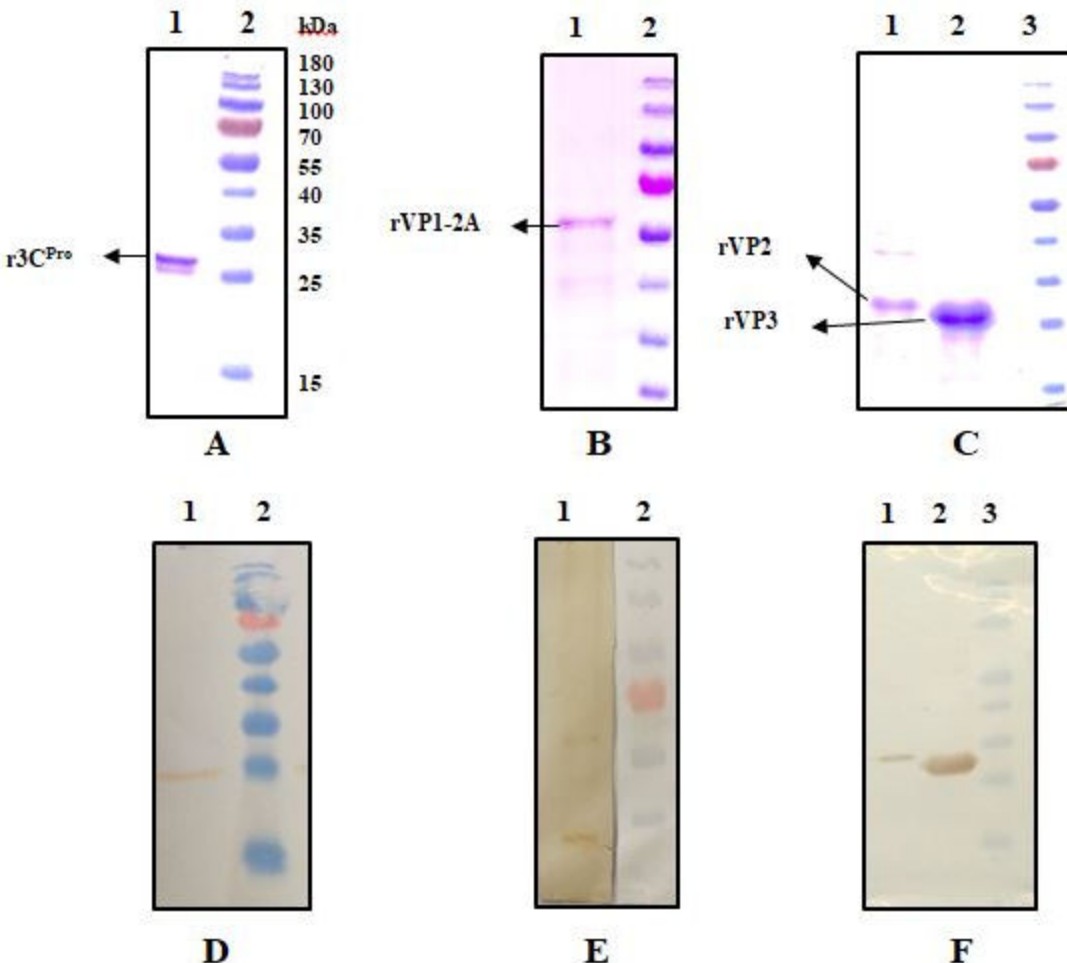

**FIG 1** SDS-PAGE and western blot analysis of HAV recombinant proteins. Coomassie brilliant blue R-250 staining: (A) r3C protein, (B) rVP1-2A protein, (C) rVP2 and rVP3 proteins. Western blot analysis with anti-His antibodies: (D) r3C$^{Pro}$, (E) rVP1-2A, (F) rVP2 and rVP3.

optimized before proceeding with the estimation of sensitivity and specificity for each assay. Table 2 lists the optimum conditions determined for each assay using checkerboard titrations of the antigen and enzyme conjugates. As expected, the amount of antigen required for coating wells depended on the purity level of HAV antigens. The pure antigens required comparatively less amount to achieve higher sensitivity and specificity for the assay.

## The anti-HAV IgG and IgM ELISA cutoff estimation

The ELISA conditions listed in Table 2 were used for screening panels of known anti-HAV IgM and IgG antibody positive and negative samples in indirect ELISAs using different HAV antigens. For each antigen, the cutoff value for anti-HAV IgM or IgG ELISA was estimated by doing ROC curve analysis by screening known anti-HAV IgG and IgM negative human serum samples (panel 1, $n = 100$), anti-HAV IgG positive serum samples (panel 2b, $n = 33$), and IgM positive serum samples (panel 3, $n = 39$) (Fig. 2). Table 3 lists the estimated values for sensitivity, specificity, the area under the curve (AUC), and the cutoff for each assay.

The two kits used for testing of serum samples to prepare anti-HAV IgM (Wantai, China) and anti-HAV IgG (GBC, Taiwan) negative and positive serum sample panels utilize tissue culture-grown whole virus as HAV antigen, which is in particulate form and highly organized rigid structure. Testing of the samples from these panels in individual HAV

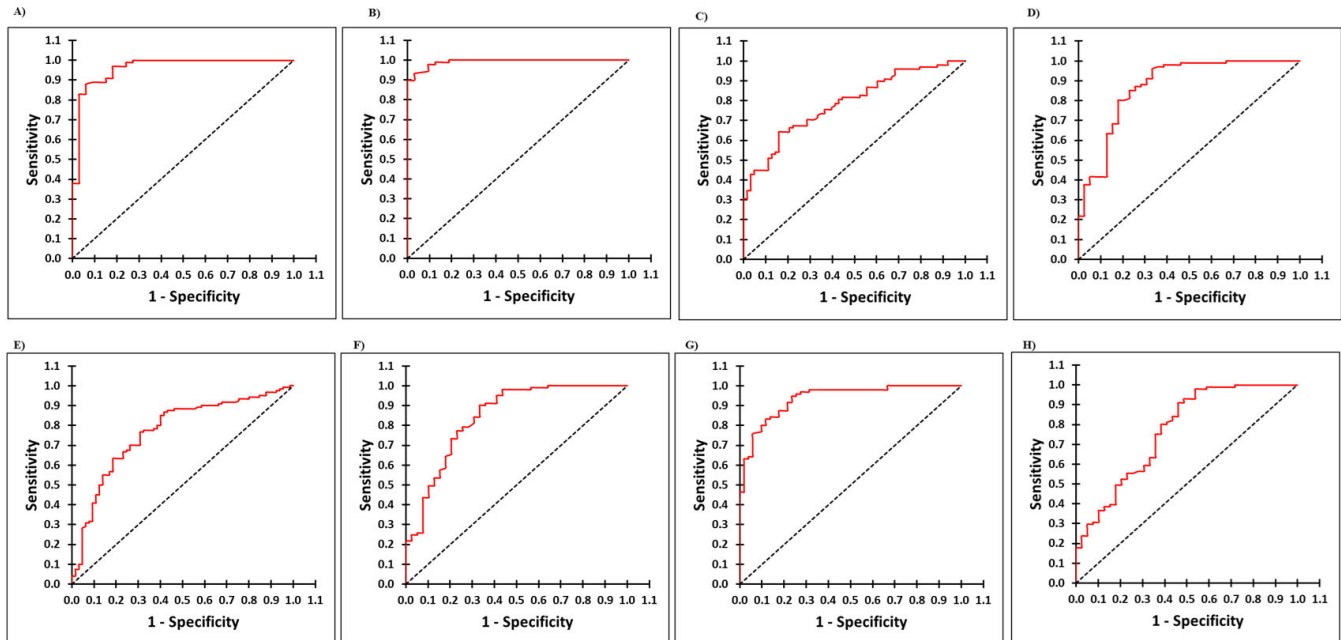

**FIG 2** ROC curve analysis for estimation of cutoff, sensitivity, and specificity for HAV recombinant protein-based indirect ELISAs to detect anti-HAV IgG and IgM antibodies: (A) anti-HAV IgG 3C$^{Pro}$, (B) anti-HAV IgM 3C$^{Pro}$, (C) anti-HAV IgG VP0, (D) anti-HAV IgM VP0, (E) anti-HAV IgG VP3, (F) anti-HAV IgM VP3, (G) anti-HAV IgG VP1-2A, and (H) anti-HAV IgM VP1-2A.

recombinant protein-based assays showed overall less sensitivity of detection for both anti-HAV IgM and IgG antibodies (Table 3). Among the three capsid proteins, rVP1-2A protein showed the highest sensitivity (86.3%) and specificity (84.2%) in detecting anti-HAV IgG antibodies in individuals having past exposure to the virus. The accuracy of this assay estimated by the ROC curve appeared to be fair (AUC = 0.935). Comparatively, VP0 and VP3 proteins showed lower sensitivities (71.4% and 69.2%, respectively) and specificities (70.4% and 71.7%, respectively) in detecting anti-HAV IgG positive samples (Table 3).

The hepatitis A virus infection is diagnosed by detecting anti-HAV IgM antibodies in symptomatic individuals and thus considered as a primary acute-phase marker (12). We evaluated all four recombinant HAV proteins generated in the study using anti-HAV IgM negative and positive serum panels (panels 1 and 3) for the possible use in HAV diagnosis in an indirect ELISA format. Anti-HAV IgM reactivity of the three capsid proteins differed significantly (Table 3). Among these, rVP0 (VP2 + VP4) exhibited the highest sensitivity (79.5%) and specificity (80.2%) with AUC = 0.80 (Fig. 2D), followed by rVP3 (Fig. 2F). Surprisingly, rVP1-2A showed the least sensitivity (64.1%) and specificity (64.4%) in detecting anti-HAV IgM positivity (Fig. 2H). Notably, rVP1-2A exhibited the highest reactivity among the capsid proteins (sensitivity: 86.3%, specificity: 84.2%) in the anti-HAV IgG assay (Fig. 2G). As compared to the capsid proteins, the r3C$^{Pro}$ protein-based IgM assay exhibited higher sensitivity (96.9%) as well as specificity (93.2%) with AUC = 0.991 (panel 3), in detecting IgM positive individuals in the acute phase of disease

**TABLE 3** Estimated cutoff values for different ELISAs

| | Protein | | | | | |
| | rVP1-2A | | rVP0 | | rVP3 | |
| ELISA | IgG | IgM | IgG | IgM | IgG | IgM |
|---|---|---|---|---|---|---|
| Sensitivity | 86.3 | 64.1 | 71.4 | 79.5 | 69.2 | 76.9 |
| Specificity | 84.2 | 64.4 | 70.4 | 80.2 | 71.7 | 76.2 |
| AUC | 0.935 | 0.769 | 0.708 | 0.8 | 0.77 | 0.844 |
| Cutoff | 0.28 | 0.21 | 0.35 | 0.22 | 0.32 | 0.29 |

(Fig. 2A). Testing of these acute-phase sample panels in the r3C$^{Pro}$ IgG assay showed 93.94% sensitivity and 88% specificity with AUC = 0.961 (Fig. 2B), indicating that r3C$^{Pro}$ is a better antigen to detect both anti-HAV IgM and IgG antibodies during the acute phase of infection.

Individuals develop antibodies against both structural (capsid proteins) and non-structural proteins of HAV when infected with the virus. In contrast, vaccinated individuals receiving attenuated HAV vaccine are known to elicit antibodies primarily against capsid proteins and low levels or no antibodies against non-structural proteins (13). To evaluate possible use of r3C$^{Pro}$ protein-based assay for differentiation between infection and vaccination, we screened 30 paired samples from vaccinated children who were anti-HAV IgG negative prior to vaccination and became anti-HAV IgG positive (tested with commercial kit) after receiving two doses of the whole virus-based inactivated vaccine (HAVRIX, GlaxoSmithKline Inc.). Anti-3C$^{Pro}$ IgG antibodies remained undetectable in 27/30 children after vaccination, while three children tested positive (Fig. 3). Thus, it appeared that practically, it would be possible to discriminate between antibodies acquired in response to natural virus infection and vaccination using 3C$^{Pro}$-based IgG ELISA. However, this would depend upon the duration of persistence of anti-3C$^{Pro}$ IgG antibodies in individuals who had past exposure to HAV.

To understand the longevity of anti-3C$^{Pro}$ antibodies after natural HAV infection, anti-HAV IgG positive samples from different age groups, 6–10 years, 15–25 years, and 40+ years (panel 2b, positive for anti-HAV IgG antibody) were tested for anti-3C$^{Pro}$ IgG antibodies with the assumption that antibodies against NSPs may not remain detectable for a long duration after primary exposure to the virus. Thus, younger children will have higher anti-NSP positivity since they would have comparatively recent virus exposure, while the positivity would subsequently decline in the older age groups, with the assumption that these individuals had a single HAV exposure in their lifetime (Fig. 4). As expected, the highest anti-3C$^{Pro}$ antibody positivity (40%) was seen in the 6–10-year age group, followed by 16.67% in the 15–25-year age group. Surprisingly, the 40+ age group showed a significantly higher positivity (26.67%) than the 15–25-year age group (Fig. 4). It appeared that at least a few of the individuals in this group had multiple virus exposures. To see whether any of these individuals testing positive for anti-3C$^{Pro}$ IgG

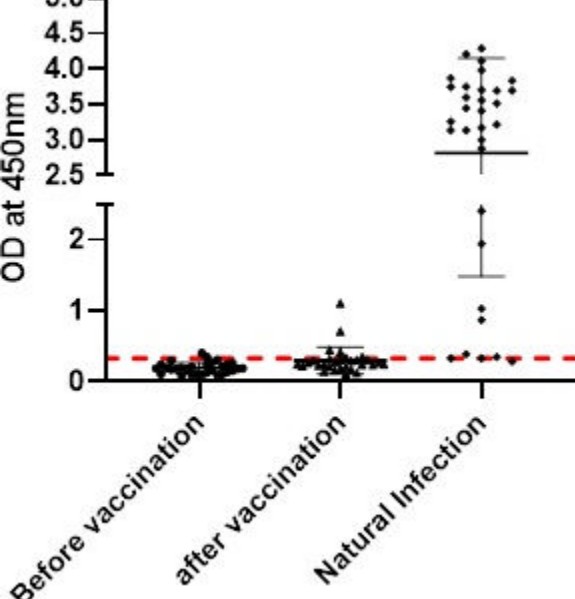

**FIG 3** 3C$^{Pro}$ IgG ELISA. Three sample panels were tested: paired samples from children pre-vaccination and post-vaccination, and acute-phase anti-HAV IgM samples (natural infection).

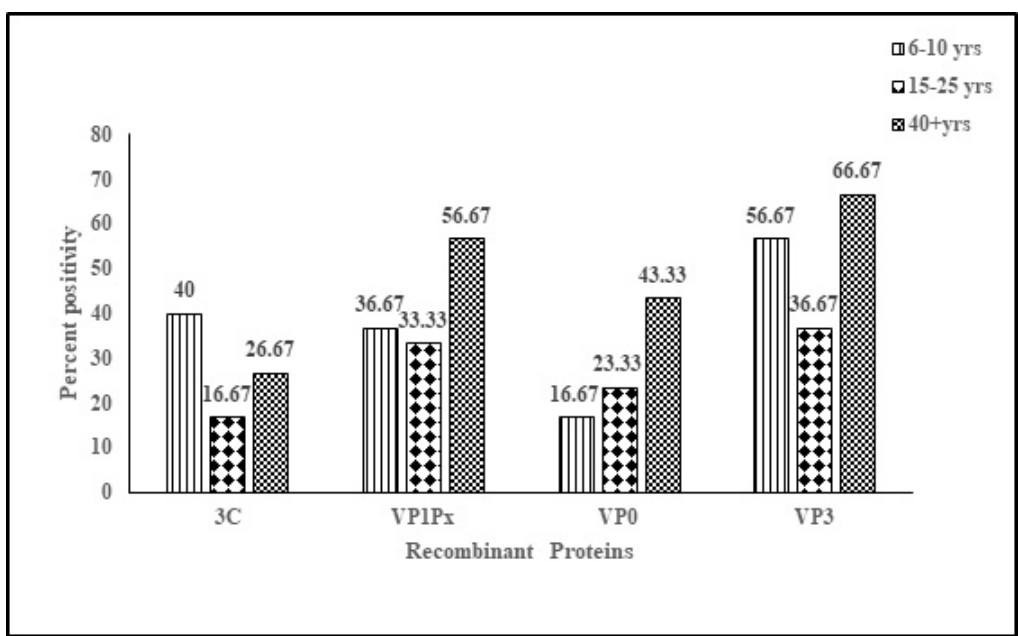

**FIG 4** Anti-HAV IgG reactivity of individual recombinant proteins in three age groups.

has a recent HAV infection, these were tested for anti-HAV IgM using a commercial kit (Wantai, China). Interestingly, 2/12 children from 6 to 10 years age group tested positive for IgM indicating recent HAV infection, while all other individuals tested negative and hence ruled out recent/current virus exposure. This panel was constituted using samples collected during a serosurvey, and it was noted that the two HAV IgM positive children were asymptomatic cases of hepatitis A. The higher anti-3C$^{Pro}$ IgG positivity in the 40+ age group was probably due to reinfection as HAV is endemic in India. These findings clearly showed that anti-3C$^{Pro}$ IgG antibody may not be a useful marker to distinguish natural infection from vaccination, as these antibodies are not persistent.

These results prompted us to look for the persistence of IgG antibodies against the three capsid proteins in different age groups (Fig. 4). Interestingly, the 6–10-year age group showed a positivity of 56.67% against VP3 protein, followed by VP1-2A (36.67%) and VP0 protein (16.67%). The 15–25-year age group showed 36.67%, 33.33%, and 23.33% positivity against VP3, VP1-2A, and VP0 proteins, respectively. Meanwhile, the 40+ age group showed 66.67%, 56.67%, and 43.33% reactivity against VP3, VP1-2A, and VP0 proteins, respectively. In all age groups, VP3 reactivity was noted to be the highest, while that for VP0 was the lowest. Notably, though all samples were anti-HAV IgG positive, as tested with whole virus antigen-based tests, reactivity with individual capsid proteins was comparatively low.

## Antigenic cross-reactivity of HAV recombinant antigens with other viruses

Patients in the acute phase of viral hepatitis are presented with similar clinical features and laboratory markers; hence, it is difficult to differentiate HAV infection from other hepatitis virus infections. Thus, the use of virus-specific diagnostic tests is required. To check whether HAV recombinant antigens generated in the present study have any cross-reactivity with other hepatitis viruses such as HEV, HBV, and HCV as well as with other picornavirus such as coxsackie A16 virus, serum samples positive for IgM antibodies against HEV, HBV, and viral RNA positive coxsackie A16 samples were tested for IgM antibodies using different antigens. For hepatitis C, HCV RNA positive (anti-HCV IgG positive) serum samples were tested for IgG antibodies using different antigens. All samples tested negative in 3C$^{Pro}$ IgM ELISA except for the 1/30 anti-HEV IgM positive sample. When re-tested with anti-HAV IgM assay, this sample tested positive for anti-HAV

IgM, confirming it to be a dual infection with HAV and HEV. All HCV IgG positive samples tested negative in IgG ELISAs, indicating that none of the HAV recombinant HAV proteins had any cross-reactivity with other hepatitis/enteric viruses.

## DISCUSSION

The major neutralization epitope/s of HAV are non-linear and highly antigenic. It requires assembly of capsid proteins so as to bring residues that are far away from each other to spatial proximity via polypeptide folding (14). However, these capsid proteins also harbor several linear antigenic domains that contribute to humoral anti-HAV response (5). It has been shown that individual capsid proteins (purified from tissue culture-grown virus) exhibit reactivity with IgM antibodies. Immunoblot analysis of HAV capsid proteins with acute and convalescent phase sera has shown that the IgM antibodies preferentially recognize the structural viral proteins VP0 and VP3, whereas IgA and IgG antibodies react more strongly with the VP1 (15). A similar analysis carried out in another laboratory has challenged these findings by documenting higher reactivity of the serum samples in the acute phase of hepatitis A against VP1 (98% IgM, 94% IgG), followed by VP0 (35% IgM, 88% IgG) and VP3 (29% IgM, 73% IgG); while individuals with past exposure to HAV have comparatively higher reactivity with VP3 (73%) than VP1 (29%) and VP0 (29%) (7). Such discrepant findings indicate that anti-HAV antibody signatures during the acute phase and after recovery are not the same.

In the present study, we evaluated the reactivity of three HAV capsid proteins, rVP1-2A, rVP0 (VP2 + VP4), and rVP3 as coating antigens in indirect ELISA format for their reactivity with anti-HAV IgM or IgG antibodies in acute phase and individuals having past exposure to HAV. It was observed that all capsid proteins reacted well with acute-phase serum samples confirming the presence of highly immunogenic linear epitopes in capsid proteins and the generation of significant antibody response against these epitopes upon HAV infection. Among these proteins, marginally higher reactivity was seen against rVP0 (sensitivity 79.5% and specificity 80%) followed by that against rVP3 (sensitivity 76.9% and specificity 76.2%) and rVP1-2A (sensitivity 64.1% and specificity 64.4%). These results were in agreement with the findings reported by Gauss-Müller and Deinhardt (15) but disagreed with the findings of Wang et al. (7), wherein they report the highest reactivity of IgM antibodies with rVP1 (98%) and very low reactivity with rVP0 (35%) and rVP3 (29%) proteins. It was likely that this difference was observed as ELISA is a more sensitive assay as compared to immunoblot assay for detection of antibodies. Nevertheless, it is important to note that in the present study, none of the recombinant capsid proteins exhibited acceptable sensitivity and specificity in ELISA for the detection of anti-HAV IgM antibodies so as to replace the whole virus antigen.

Several studies have suggested VP1 to be an immunodominant structural protein of HAV (16–18). Most of these studies have used serum samples collected at one time point from infected individuals. It appears that the duration between sample collection and primary virus exposure could be a crucial factor. Furthermore, antibodies developed against linear epitopes of different capsid proteins may have different half-lives. Our assessment of the persistence of antibodies against individual capsid proteins after natural infection (anti-HAV IgG positive) in different age groups also showed that the reactivity is not uniform in all age groups. It was noted to be higher for rVP1-2A and rVP3 in the 6–10-year and 40+ age groups. A lifelong persistence of anti-HAV antibodies is known, but these assessments are mainly done by using whole virus-based commercial assays. It would require a well-planned long-term follow-up study for reliable assessment of antibody response against individual structural proteins of HAV.

We further evaluated r3C$^{Pro}$ for differentiation between humoral response conferred by natural infection and that by immunization using defined serum panels. HAV 3C$^{Pro}$ is the marker of active viral replication after infection with HAV. For determining longevity of the anti-3C$^{Pro}$ response after natural HAV infection, anti-HAV IgG positive individuals (having past exposure to the virus) from different age groups were screened. Since India is endemic to HAV, children are exposed to the virus

during early childhood with nearly 100% seroconversion by the age of 10 years. As expected (due to recent exposure), the 6–10-year age group showed relatively higher anti-3C$^{Pro}$ antibody positivity (40%), followed by the 15–25-year group (16.67%). Surprisingly, the 40+ age group showed comparatively higher positivity than the 15–25-year age group, probably due to sporadic re-infection with HAV. It was likely that though primary exposure to HAV provides lifelong protection because of antibodies produced by long-lived plasma cells, re-exposure to the virus resulted in the recall response due to activation of memory B cells, resulting in the enhancement of antibody levels. Overall, these results showed that there is waning of anti-3C IgG antibodies in individuals who were infected with the virus, though total anti-HAV IgG antibodies remain detectable almost lifelong.

Persistence of anti-3C$^{Pro}$ IgG antibody until 105 weeks (735 days) in experimentally infected chimpanzees (19) and 15 months post-infection in children (10) has been reported. These reports indicate that anti-3C$^{Pro}$ IgG antibodies may persist until ~ 2 years post-exposure. The presence of anti-3C$^{Pro}$ IgG antibodies in 3/30 vaccinated individuals and the absence in 27/30 individuals suggested possible utility of anti-3C$^{Pro}$ IgG antibody assay to differentiate between infection and vaccination in children. Three vaccinated children testing anti-3C$^{Pro}$ positive probably had exposure to the virus and were in the incubation phase before their pre-vaccination sample was taken. It was also likely that they were infected after taking the first dose of vaccine and before developing a protective immune response.

Although screening of vaccinated children indicated possible use of anti-3C IgG antibodies to differentiate between antibody response after natural infection and vaccination, longevity of anti-3C$^{Pro}$ antibodies for a limited duration, ~2 years post-infection, clearly ruled out their utility in differentiating antibody response developed after natural infection and after vaccination.

Khudyakov et al. (11) have demonstrated the presence of six immunoreactive peptides within 3C$^{Pro}$ protein, and one of these to be highly immunoreactive with acute-phase sera. Hence, we checked whether 3C$^{Pro}$ antigen has any utility for hepatitis A diagnosis. Interestingly, the r3C$^{Pro}$-based IgM assay exhibited 96.9% sensitivity and 93.2% specificity in detecting IgM antibodies in the acute-phase samples. Most importantly, this assay had no cross-reactivity with samples from other acute hepatitis cases diagnosed to be either hepatitis E or hepatitis B. Also, no cross-reactivity was seen with serum samples from individuals who were HCV and Hand, Foot, and Mouth Disease virus RNA positive. Thus, 3C$^{Pro}$ seemed to be the best candidate antigen for the development of hepatitis A diagnostic assay.

In conclusion, though recombinant HAV capsid proteins harbor linear epitopes and are immunoreactive, these cannot be used as a replacement for whole virus as antigen. However, non-structural proteins such as 3C$^{Pro}$ appear to be a promising candidate antigen for use in hepatitis A diagnostic assays to detect anti-HAV IgM antibodies. We feel that blocking of 3C$^{Pro}$ reactive IgG antibodies may further improve sensitivity and specificity of the assay. The development of such an assay in a rapid immunochromatographic format would be very useful for point-of-care diagnosis.

## ACKNOWLEDGMENTS

This study was funded by ICMR-National Institute of Virology intramural funds.

K.L. conceived the concept. S.H. and N.G. conducted laboratory work. A.W. performed the statistical analyses. K.L. and S.H. analyzed the data and interpreted the results. K.L. and S.H. wrote the manuscript.

## AUTHOR AFFILIATION

[1]ICMR-National Institute of Virology, Pune, Maharashtra, India

## AUTHOR ORCIDs

Supriya Hunderkar http://orcid.org/0009-0004-2122-4414
Kavita Lole http://orcid.org/0000-0001-6131-2268

## FUNDING

| Funder | Grant(s) | Author(s) |
| --- | --- | --- |
| Indian Council of Medical Research (ICMR) | Indirect funding | Kavita Lole |

## AUTHOR CONTRIBUTIONS

Supriya Hunderkar, Investigation, Methodology, Validation, Writing – original draft, Writing – review and editing | Nital Ganorkar, Investigation, Methodology | Atul Walimbe, Data curation, Formal analysis | Kavita Lole, Conceptualization, Formal analysis, Funding acquisition, Methodology, Project administration, Supervision, Writing – review and editing

## ETHICS APPROVAL

Institutional ethics committee approval (NIV/IEC/Jan/2020/D-21) was obtained for the use of human serum samples in the study.

## ADDITIONAL FILES

The following material is available online.

Open Peer Review

**PEER REVIEW HISTORY (review-history.pdf).** An accounting of the reviewer comments and feedback.

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
