## [Reviewer comments · Microbiology Spectrum]

Microbiology Spectrum

Evaluation of hepatitis A virus recombinant proteins for detecting anti-HAV IgM and IgG antibodies

Supriya Hunderkar, Nital Ganorkar, Atul Walimbe, and Kavita Lole

Corresponding Author(s): Kavita Lole, National Institute of Virology

Review Timeline:

Submission Date:	June 22, 2024
Editorial Decision:	August 12, 2024
Revision Received:	October 9, 2024
Accepted:	January 25, 2025

Editor: Saumitra Das

Reviewer(s): The reviewers have opted to remain anonymous.

Transaction Report:

DOI: <https://doi.org/10.1128/spectrum.01528-24>

Re: Spectrum01528-24 (Evaluation of hepatitis A virus recombinant proteins for detecting anti-HAV IgM and IgG antibodies)

Dear Dr. Kavita S Lole:

Thank you for the privilege of reviewing your work. Below you will find my comments, instructions from the Spectrum editorial office, and the reviewer comments. Please address the comments appropriately.

Revision Guidelines

Sincerely,
Saumitra Das
Editor
Microbiology Spectrum

Reviewer #1 (Comments for the Author):

- 1) A more detailed figure legend corresponding to figure 2 would be helpful for the general reader.
- 2) In the para covering lines 264-278, indication of specific figure or table is missing and only panel numbers are indicated.
- 3) In lines 283-285, the authors have tested only 30 serum samples. The statistical tool that was used to arrive at the number of samples required for this experiment and the subsequent inference drawn should be indicated.

Reviewer #2 (Comments for the Author):

The current study by Hundekar S. et. al entitled, "Evaluation of Hepatitis A virus recombinant proteins for detecting anti-HAV IgM and IgG antibodies" presents a comprehensive case for prospective Hepatitis A diagnosis kits using antigen capture ELISA using recombinant structural and non-structural proteins of Hepatitis A.

The manuscript is interesting, well-written, experiments are designed properly, however there are few critical issues, which should be clarified before consideration:

Major Comments-

1. The study highlights that even though there are strong antigenic epitopes in the structural (capsid protein majorly) and non-structural protein (3CPro) but they are unable to match the sensitivity and specificity of conventional whole antigen ELISA kits available for detection of anti-HAV IgM/IgG. In this context, what is the rationale/advantage behind the submitted work?
2. As reported by authors, the non-structural protein (3CPro) showed higher sensitivity and specificity as compared to structural recombinant proteins. This study also finds that the anti-3CPro antibodies are short lived, mostly present in acute phases of infection, even though there are reports that anti-3CPro antibodies persisted in chimpanzees and 15 months post-infection children for as long as 2 years post exposure. This ambiguity and subsequent suitability of using this protein needs further exploration.
3. Known IgG positive samples of different age groups, tested for persistence of anti-3CPro IgG antibodies, showed a slightly skewed trend, 40+ age group samples showed significant positivity which is contrary to the assumption and makes the proposed diagnostic marker unsuitable for endemic areas and also cannot discriminate between natural infection and vaccination.
4. Figures are not represented with standard deviation and level of significance.
5. Overall conclusion of the manuscript is not very clear, interpretations are very subjective and the importance of the study is not projected properly.

Comments:

1) A more detailed figure legend corresponding to figure 2 would be helpful for the general reader.

2) In the para covering lines 264-278, indication of specific figure or table is missing and only panel numbers are indicated.

3) In lines 283-285, the authors have tested only 30 serum samples. The statistical tool that was used to arrive at the number of samples required for this experiment and the subsequent inference drawn should be indicated.

The current study by Hundekar S. et. al entitled, “**Evaluation of Hepatitis A virus recombinant proteins for detecting anti-HAV IgM and IgG antibodies**” presents a comprehensive case for prospective Hepatitis A diagnosis kits using antigen capture ELISA using recombinant structural and non-structural proteins of Hepatitis A.

The manuscript is interesting, well-written, experiments are designed properly, however there are few critical issues, which should be clarified before consideration:

Major Comments-

1. The study highlights that even though there are strong antigenic epitopes in the structural (capsid protein majorly) and non-structural protein (3C^{Pro}) but they are unable to match the sensitivity and specificity of conventional whole antigen ELISA kits available for detection of anti-HAV IgM/IgG. In this context, what is the rationale/advantage behind the submitted work?
2. As reported by authors, the non-structural protein (3C^{Pro}) showed higher sensitivity and specificity as compared to structural recombinant proteins. This study also finds that the anti-3C^{Pro} antibodies are short lived, mostly present in acute phases of infection, even though there are reports that anti-3C^{Pro} antibodies persisted in chimpanzees and 15 months post-infection children for as long as 2 years post exposure. This ambiguity and subsequent suitability of using this protein needs further exploration.
3. Known IgG positive samples of different age groups, tested for persistence of anti-3C^{Pro} IgG antibodies, showed a slightly skewed trend, 40+ age group samples showed significant positivity which is contrary to the assumption and makes the proposed diagnostic marker unsuitable for endemic areas and also cannot discriminate between natural infection and vaccination.
4. Figures are not represented with standard deviation and level of significance.
5. Overall conclusion of the manuscript is not very clear, interpretations are very subjective and the importance of the study is not projected properly.

Response to reviewers:

Reviewer 1:

- 1) In the para covering lines 264-278, indication of specific figure or table is missing and only panel numbers are indicated.

Response: We are extremely sorry for this inadvertent mistake. These details have been mentioned in the text now. Additions are highlighted in red.

- 2) In lines 283-285, the authors have tested only 30 serum samples. The statistical tool that was used to arrive at the number of samples required for this experiment and the subsequent inference drawn should be indicated.

Response: These samples were from an outbreak of hepatitis A, investigated by ICMR-National Institute of Virology. These were archived paired samples from children who were anti-HAV IgG negative and hence immediately immunized to protect them. No statistical analysis was done to decide this sample number.

- 3) The study highlights that even though there are strong antigenic epitopes in the structural (capsid protein majorly) and non-structural protein (3C^{Pro}) but they are unable to match the sensitivity and specificity of conventional whole antigen ELISA kits available for detection of anti-HAV IgM/IgG. In this context, what is the rationale/advantage behind the submitted work?

Response: Several studies have shown limitations of recombinant HAV capsid proteins for hepatitis A diagnosis, however, these conclusions were based on a few serum samples from anti-HAV positive humans or experimentally infected animals by estimating sensitivity and specificity. In the present study, we have used well defined human serum panels from acute phase patients, individuals having past HAV exposure and vaccinated individuals for the assessment of reactivity of different HAV recombinant antigens.

There are reports stating use of anti-3C IgG antibodies as marker for differentiating antibody response due to natural infection and vaccination (Kabrane-Lazizi et al., 2001; Stewart et al., 1997). Conversely, our age-wise screening of individuals having past HAV exposure shows that anti-3C antibody response is short lived and hence cannot be used for this purpose. We also screened acute phase hepatitis A patients for anti-3C IgM antibodies and noted that this antigen has comparable sensitivity and specificity to whole virus based antigen in picking up true positive cases. Most importantly, 3C^{Pro} protein based assay has never been evaluated as a

hepatitis A diagnostic assay (anti-HAV IgM detection). This is the novel finding from this study.

Lines 396-398 and 408-411 have been added in the discussion of manuscript to have better clarity.

- 4) As reported by authors, the non-structural protein (3C^{Pro}) showed higher sensitivity and specificity as compared to structural recombinant proteins. This study also finds that the anti-3C^{Pro} antibodies are short lived, mostly present in acute phases of infection, even though there are reports that anti-3C^{Pro} antibodies persisted in chimpanzees and 15 months post-infection children for as long as 2 years post exposure. This ambiguity and subsequent suitability of using this protein needs further exploration.

Response: Hepatitis A diagnosis is done by detecting anti-HAV IgM antibodies. Commercial assays use whole virus as the detecting antigen in these assays. Although individual capsid proteins harbour linear B-cell epitopes, they show comparatively lower sensitivity in detecting anti-HAV antibodies. Importantly, HAV non-structural proteins also elicit immune response. Our results show that though antibodies against 3C^{Pro}, a non-structural protein, are short lived, anti-3C IgM antibodies have potential to be used for diagnosis of HAV infection. A previous study by Stewart et al (1997) has demonstrated persistence of anti-3C antibodies in two experimentally infected Chimpanzees for up to 105 weeks, while a study by Kabrane-Lazizi et al (2001) demonstrated its persistence for up to 15th months in infected children. In our study, we checked anti-3C antibodies using two sample sets-

Set I (individuals of different age groups having past HAV exposure, i.e. positive for anti-HAV IgG but negative for IgM): checked for anti-3C IgG antibodies

Set II (confirmed acute phase hepatitis A cases testing IgM antibody positive): checked for anti-3C IgM antibodies.

The screening of acute phase patient samples clearly showed that anti-3C IgM antibody could be used for diagnosis of hepatitis A. Most importantly, this assay did not show any cross-reactivity with other hepatitis viruses.

- 5) Known IgG positive samples of different age groups, tested for persistence of anti-3C^{Pro} IgG antibodies, showed a slightly skewed trend, 40+ age group samples showed significant positivity which is contrary to the assumption and makes the proposed diagnostic marker unsuitable for endemic areas and also cannot discriminate between natural infection and vaccination.

Response: We are proposing use of anti-3C IgM antibody detection as the diagnostic test for hepatitis A and not IgG antibody detection. The age group wise screening of positivity for anti-3C antibodies involved detection of IgG antibodies which represented past exposure to the virus.

- 6) Overall conclusion of the manuscript is not very clear, interpretations are very subjective and the importance of the study is not projected properly.

Response: The answer will be same as that for the Q3 and Q4.

Re: Spectrum01528-24R1 (Evaluation of hepatitis A virus recombinant proteins for detecting anti-HAV IgM and IgG antibodies)

Dear Dr. Kavita,

Your manuscript has been accepted, and I am forwarding it to the ASM production staff for publication. Your paper will first be checked to make sure all elements meet the technical requirements. ASM staff will contact you if anything needs to be revised before copyediting and production can begin. Otherwise, you will be notified when your proofs are ready to be viewed.

Sincerely,
Saumitra Das
Editor
Microbiology Spectrum

Reviewer (Comments for the Author):

Satisfactory response has been provided point by point.

Satisfactory point by point responses have been provided by the author.